# The Health Effects of a Cherokee Grounded Culture and Leadership Program

**DOI:** 10.3390/ijerph19138018

**Published:** 2022-06-30

**Authors:** Melissa E. Lewis, Jamie Smith, Sky Wildcat, Amber Anderson, Melissa L. Walls

**Affiliations:** 1Department of Family and Community Medicine, School of Medicine, University of Missouri, Columbia, MO 65212, USA; smithjami@health.missouri.edu; 2Department of Rehabilitation, Human Resources, & Communication Disorders, University of Arkansas, Fayetteville, AR 72701, USA; skybwildcat@gmail.com; 3Department of Biostatistics and Epidemiology, Hudson College of Public Health, University of Oklahoma Health Sciences Center, Oklahoma City, OK 73104, USA; amber-s-anderson@ouhsc.edu; 4Center for American Indian Health, Bloomberg School of Public Health, Johns Hopkins University, Duluth, MN 55812, USA; mwalls3@jhu.edu

**Keywords:** Indigenous youth, culture as prevention, Indigenous mental health

## Abstract

Introduction: Indigenous youth and young adults endure some of the highest rates of physical and mental health problems in the United States compared to their non-Indigenous counterparts. Colonization, oppression, and discrimination play a substantial role in these inequitable disease rates. However, culture (e.g., identity, participation, and connection) relates to the prevention of and recovery from illness in Indigenous populations. The Remember the Removal program aims to teach Indigenous youth and young adults tribally specific culture, history, and language to put them on a trajectory to become informed and culturally connected community leaders. We examined the program’s effects on health. Method: Thirty Remember the Removal program participants, mainly young adults, completed surveys four times: before the program’s start, at the end of the training period, at the program’s end, and at a six-month follow-up. Various indicators of physical, mental, spiritual, and cultural health and well-being were measured at each time period. Paired *t*-tests were completed to compare baseline scores to each subsequent time interval. Results: At program completion, and as indicated with an asterisk at the six month follow-up, participants had statistically significantly improved diet and exercise measures (e.g., reduced sugary, salty, and fatty foods, reduced soda consumption, increased fruit consumption, and improved self-efficacy for exercise), improved mental health indicators (e.g., reduced stress, anxiety, depression*, anger*, post-traumatic stress disorder, and microaggressions*, and improved positive mental health) and improved social and cultural connection (e.g., social support, Cherokee identity*, Cherokee values). Discussion: This is one of the first quantitative studies to demonstrate the profound effects that cultural learning and connection have on the health and well-being of Indigenous people and practices. It also demonstrates the specificity and effectiveness of a program created by and for tribal citizens. Future programs with Indigenous populations should work to center cultural connection and ensure that programs are created and directed by tribal community members.

## 1. Introduction

Indigenous youth and adults face some of the worst health conditions in the United States including higher rates of depression, anxiety, post-traumatic stress disorder, obesity, diabetes, and cardiovascular disease compared to their non-Indigenous counterparts [1,2,3,4,5,6]. Unfortunately, many efforts to address these inequities have included programs that are not developed for or by Indigenous communities with their health contexts or needs in mind resulting in nonsignificant results, high attrition rates, low sustainability, and/or no long-term improvement [7,8,9,10]. Cultural relevance is a critical part of any health intervention: when health interventions are designed for a specific cultural or ethnic group, health improvements increase four-fold [11]. Unfortunately, very few studies are created by tribal communities themselves using their own values and beliefs to guide the intervention.

Health inequity arises from the multiple social and economic inequities that Indigenous people experience [12], including underfunded hospitals [13] and schools [14], discrimination and racism around housing [15] and job opportunities [16], mascotry or invisibility (erasure) in society [17], and unaddressed colonial practices and the subsequent trauma [18] they cause. In all these ways, colonization continues to play a subversive role in the United States that permeates the lives of Indigenous people, reducing health and well-being outcomes [19].

To counter the negative and pervasive effects of colonization and regain balance, tribes and communities have led efforts to revitalize their cultural and spiritual ways of life [20]. These efforts include teaching children and families crafts, history, stories, values, foodways, language, and ceremonies [21,22,23,24,25,26,27]. Indigenous people note that they feel more connected and healthier when they participate in their cultural ways of life [28]. Culture appears to mediate the health risks that colonization has created. Evidence-based interventions that are tailored to Indigenous people are more effective than those without tailoring [29,30]. Programs that are created from tribal communities using Indigenous knowledge solely have also seen great success [31,32,33].

One program, the Remember the Removal program, began in 1984 [34] by a grassroots group of Cherokees who wanted to ensure that their youth would not succumb to alcohol and drugs and instead become culturally informed and confident citizens that could use their Cherokee values to become future leaders within their families and communities. This group decided that the best way to reach this goal was to teach the youth Cherokee culture, history, language, and values through a program in which they would ride their bicycles 1000 miles across the route in which their ancestors were forcibly removed by the United States government 250 years earlier. Participants met 2–3 times a week from January through May and receive training in Cherokee language, culture, history, genealogy, and bicycle training. Classes are in-person and include classroom lectures, readings, and site visits. A Cherokee genealogist completed a family tree and report for each participant and then a Cherokee educator taught Cherokee kinship terms and values. In May, participants depart for three weeks and follow the Northern removal route on bicycles. They visit historical and cultural sites facilitated by Cherokee historians and cultural experts; they read historical, geo-located primary materials about the removal, and receive on-site, land-based cultural and historical education (see [35] for additional details on cultural training). This program continues today given its formidable reputation and support within the Cherokee community and has expanded to the Eastern Band of Cherokee Indians as well.

We set out to assess the effect of the program on participants starting in 2015 given the legacy of community leaders that have been created through this program. While we are not administrators of the program, we collaborated with program staff to accomplish this goal of program assessment. Focus group results from the 1984 and the 2015 cohorts of the RTR program indicate that after this program, Cherokee youth feel more confident, knowledgeable, and connected to their Cherokee identity, have improved health behaviors, and are more involved in their tribal community [34]. These results were sustained for over thirty years. Given these impressive results coupled with community reports, further inquiry was carried out. Based on focus group themes, a survey was created to test the change in participants’ health and well-being. Specifically, the purpose of this study was to describe changes in RTR participants’ physical, emotional, social, and cultural health and well-being before, during, at program completion, and six months after the completion of the program.

## 2. Methods

The Cherokee Nation (CN) and Eastern Band of Cherokee Indians (EBCI) participate in the Remember the Removal (RTR) program yearly with slight differences between them. For instance, CN requires participants to be between the ages of 16–24 while there is no age requirement for EBCI participants; CN houses this program in the education department while the RTR program is within a diabetes prevention program at EBCI. Each tribe has unique selection criteria but generally, participants who share an interest in learning about Cherokee history and culture, as well as are committed to completing program requirements are ranked highest for program selection. A complete description of this program is detailed in a previous manuscript [35]. For this assessment project, we collected data at each of these time points: (1) Baseline: January, (2) Completion of the training period (pre-ride): May, (3) Completion of Intervention (post-ride): June, and (4) Six-month follow-up: December.

The research project (i.e., program assessment) was a collaborative effort between the researcher who is an enrolled citizen of the Cherokee Nation, and the RTR program staff at Cherokee Nation and at the Eastern Band of Cherokee Indians. Consistent with Indigenous and community-based participatory research methods [36,37], this project was carried out after four years of collaboration including monthly meetings to ensure that the project was appropriate and beneficial to the tribe and its citizens (See Figure 1). A cyclical process of planning, implementation, and dissemination back to key stakeholders occurred several times throughout this project. The research was approved by the Cherokee Nation IRB, the Eastern Band of Cherokee Indians IRB, and the University of Missouri IRB. Per policy, all research presentations and manuscripts were approved by the Cherokee Nation Institutional Review Board (IRB), as well as, the RTR alumni group. All research presentations included RTR participants as speakers, data collection was assisted by RTR alumni, and the first manuscript was co-authored by the RTR alumni association. Three of the five authors of this manuscript are citizens of the Cherokee Nation and two are alumni of the RTR program.

## 3. Participants

The purpose of this project was to assess the health effects of a Cherokee Nation-sponsored program, Remember the Removal. Inclusion criteria included that participants were selected by their respective tribes for the program. All participants completed an informed consent document and assent documents were completed by individuals and parents under the age of 18. The PI collaborated with program staff at each site to contact RTR participants to notify them about the research opportunity. Participants completed the survey online and a computer lab was available for those who did not have access to a computer. Participants received $30 and a small gift at each survey interval (×4). Gifts included Cherokee baskets, mugs, scarves, and a backpack with the RTR logo.

There are a limited number of RTR participants selected each year from each tribe given the safety concerns that present when there are too many bicyclists on the road at the same time. Therefore, we collected data for two years aiming to satisfy the minimum needs to complete a *t*-test (*n* = 30). Therefore, there were 19 participants each year. The participant sample size at each time interval is presented in Table 1. In year 1, 16 out of 19 participants completed surveys at all waves (16% total attrition). In year 2, 14 out of 19 participants completed surveys at all waves (26% total attrition).

## 4. Measurement

Health, illness, and well-being are concepts that are embedded in a cultural context [38]. Measures were selected that reflected Western, Indigenous and Cherokee-specific realms of health and well-being. Indigenous communities are more likely to view health as interconnected domains of physical, emotional, and social health [39,40,41]. Further, after qualitative analysis, specific measures were selected that corresponded with relevant themes of RTR alumni experiences of the program. As with most qualitative work, the data was rich and we worked to select measures that corresponded with focus group themes, but we were unable to cover all themes and worked to select the most relevant.

For instance, within the physical domain, while some participants in the 1984 group no longer exercised at the same rate as they did at that time, during focus group discussions, many recounted skills around learning to exercise that are still utilized (e.g., self-efficacy for activity). Similarly, 2015 participants shared that they gained skills around shopping and consuming healthier foods during the program or reducing sugar-sweetened beverages so we ensured those measures were included [34].

Concepts around mental health that were discussed in focus groups were related to experiences of historical trauma and myriad emotional responses to it (e.g., anxiety, depression, stress, anger) [42]. Further, experiences of bias, racism, and microaggressions were shared, including the ways that individuals dealt with them throughout the program, and we worked to include these experiences within the survey measures (e.g., microaggressions). Accounts of flourishing mental health and resilience were plentiful in focus groups accounts and so we ensured that this was measured as well (e.g., positive mental health). These accounts of positive experiences were strongly tied to positive relationships and culture [35]. Within focus groups, participants recounted strong relationships between one another enduring over 30 years, improved relationships with family members when they returned home, an increased connection to their tribe and community, and increased Cherokee identity. These specific experiences are represented within the measures of the social and cultural domains (e.g., social support, Cherokee identity, Cherokee values). Below is a list of all measures used in this study by domain.

### 4.1. Physical Health

Body Mass Index (BMI) and Diet: Participants’ height and weight, along with age were used to calculate BMI at each wave of the study.

Food Intake: We assessed the intake of fruit, salty, sweet, and high-fat content foods, and sugary beverages by frequency based on nutritional guidelines, (e.g., How many days per week would you say you eat fruit at least 3 servings per day?) [43].

Physical Activity: The International Physical Activity Questionnaire-Short Form (IPAQ-SF) has 6 questions about different physical activities grouped into low (walking), moderate, and vigorous. The frequency of activity is gathered by week and minutes per day if applicable. For example, “During the past 7 days, did you walk for at least ten minutes for any of these reasons?”

Self-efficacy for Activity: The self-efficacy for activity scale was developed for youth to measure their perceived ability to successfully be physically active [44]. Thirteen items examine participants’ belief in their ability to complete physical activity in several scenarios with a dichotomous, yes/no, response option. For example, “I can be physically active no matter how busy my day is”.

### 4.2. Mental Health

Positive Mental Health: The mental health continuum—short form is a 14-item instrument used to assess positive mental health in terms of emotional (EWB), social (SWB), and psychological well-being (PWB) [45,46]. The items are scored on a 6-point Likert scale based on how many times they have experienced these positive feelings in the past month.

Post-traumatic Stress Disorder. The Primary Care PTSD Screen (PC-PTSD-5) is a 5-item construct designed to measure PTSD symptoms and provide a preliminary diagnosis. A sample question from this tool is, “Have you been constantly on guard, watchful, or easily startled?” When administered by trained interviewers, this construct demonstrated perfect inter-rater reliability for diagnoses and κ > 0.95 at the item level [47].

Stress: The global stress assessment is a one-item questionnaire used to assess participants’ overall stress using a 1-point scale, ranging from 1 (no stress) to 7 (extremely high stress).

Depression: The Patient Health Questionnaire-9 (PHQ-9) is a nine-item measure of the diagnostic criteria for depression [48,49,50]. Each criterion is scored based on the frequency of symptoms from 0–3, with 0 being “not at all” and 3 being “nearly every day”.

Anxiety: The Generalized Anxiety Disorder 7-item Scale (GAD-7) is a seven-item questionnaire used to identify those with GAD and to determine the severity of their anxiety [46]. The questions were selected based on anxiety symptom criteria from the Diagnostic and Statistical Manual of Mental Disorders, 4th edition (DSM-IV) [51] and are scored based on the frequency of symptoms using a 4-point Likert scale from 0–3 with 0 being “not at all” and 3 being “nearly every day”. Samples of the symptoms listed in the GAD-7 include “feeling nervous, anxious, or on edge”, “worrying too much about different things”, and “becoming easily annoyed or irritable”. The GAD-7 has demonstrated high reliability and validity [51].

Anger: Anger was measured using the tri-ethnic anger scale [52]. Questions ask about the frequency of feelings of anger from 1 (none of the time) to 3 (most of the time). We used five of the six total items for brevity and summed them for a total score.

Microaggressions: The microaggressions measure has 11 items that examine experiences of bias and discrimination that relate to the participant’s cultural group membership. We deployed the adapted version by Sittner et al. [53] that was developed from the original 33-item measure developed by Chae, Walters [54] which demonstrated good internal reliability with Cronbach’s alpha of 0.97. Items ask about the experience of discrimination including, “unfair treatment by bosses/supervisors”, “being called something racist”, and “feeling like Native people are invisible”. The items were scored on a 3-point Likert scale from 0 (this never happened), to 2 (this happened and I was bothered by it)”.

Historical Trauma. The Historical Loss and Associated Symptoms scales were developed and revised based on elders’ and tribal advisory boards’ comments and suggestions [50]. The purpose of this scale is to assess the prevalence and immediacy of thoughts pertaining to historical loss by asking “How often do you think about” and listing items such as loss of our land, loss of our language, and losing our traditional spiritual ways. Participants are asked how often they experience historical loss thoughts on 12 items on a 6-point Likert scale from 1 (never) to 6 (several times a day). This scale had a Cronbach’s alpha coefficient of 0.89 with high internal reliability [55].

### 4.3. Social/Cultural Health

Social Support: The social support survey is a 19-item measure of functional support that comprises five subscales: emotional, informational, tangible, affectionate, and positive [56]. The survey asks how often each item is available to the participant, such as “Someone you can count on to listen to you when you need to talk”. The items were ranked on a 5-point Likert scale, from 1 (none of the time) to 5 (all of the time). Reliability and validity estimates are high for this measure [56].

Cherokee Identity: The Multigroup Ethnic Identity Measure (MEIM) is a 20-item measure used to assess one’s subjective association with their ethnic group [57]. The non-specific phrase “ethnic identity” was changed to “Cherokee identity” to specify the population. It is divided into two subscales and captures attitudes regarding ethnic identity (14 questions) and other-group orientation (6 questions), i.e., understanding one’s ethnic identity and relation to other groups. The statement “I have a strong sense of belonging to my own ethnic group” exemplifies the former, and the statement “In order to learn more about my ethnic background, I have often talked to other people about my ethnic group” exemplifies the latter [58]. This instrument is scored on a 4-point Likert scale from 1 (strongly disagree”) to 4 (strongly agree). The MEIM has shown strong reliability when tested with high school and college students with Cronbach alphas of 0.81 and 0.90, respectively, and across a diversity of ethnicities [57].

Cherokee Language: One item was used to measure the frequency of use of the Cherokee language: “My ability to speak Cherokee now can be described as…” Responses were recorded on a five-point scale with zero representing ‘I cannot understand or speak’ and four representing ‘I am fluent’.

Cherokee Traditional Ways: Participants were asked “How much do you live by or follow traditional Cherokee ways?” with a 5-level Likert scale ranging from 1 (not at all) to 5 (a great deal).

Community Values: Cherokee Nation community values were measured using a 20-item instrument that was adapted for this project from a document created by Cherokee Nation regarding Cherokee-specific principles and directions around community relationships [59]. The content was created by a Cherokee elder, Benny Smith, based on his cultural and spiritual teachings. Mr. Smith was consulted and permitted the use of these values for this project. Participants responded to the statement by indicating the degree that they embodied each value using a 1 (not at all true of me) to 5 (completely true of me) Likert scale. For example, “Treat each other’s existence as being sacred or important”.

## 5. Analysis

Descriptive statistics were examined for all participant characteristics. All outcome data were examined for potential outliers. Distributions were found to be within normal symmetric boundaries and determined to be suitable for further analysis. The primary purpose of this program assessment was to analyze the change in participants from before they entered the program throughout each stage of the project to learn about patterns related to each stage of the program. Therefore, to evaluate change in participants’ physical, mental, and social/cultural health over time, paired t-tests were used to calculate mean differences in participants’ baseline scores compared to scores from each time interval (training period, intervention, and follow-up). The general ‘rule of thumb’ for a paired t-test is 30 pairs, and despite a low sample size, we were able to meet this standard in all but one analysis that had 29 matched pairs (baseline-follow-up). All analyses were performed in SAS for Windows version 9.4 (SAS Institute, Cary, NC, USA).

## 6. Results

Demographics: Two years of cohorts of RTR program participants were asked to participate in this research project. There were 38 participants overall (19 per year) with the majority being from the Cherokee Nation (63%). Women made up 58%; 36% reported having a college or advanced degree, and 89% reported living within their reservation or boundary (see Table 2). The average age of participants at baseline was 26 (Median = 21.5, SD = 11.5) with an age range of 16–61; however, only one-third of the participants were over the age of 24, and this group was primarily EBCI citizens given their inclusion requirements.

Physical Health: Participants reported higher levels of physical activity (METs) from baseline through the training period (*p* = 0.001; see Table 3) No differences among physical health behaviors, however, were noted at program completion or follow-up.

Participants reported improved eating habits during the program but not at the six month follow-up period: higher levels of fruit consumption were noted from baseline to the training period (*p* = 0.01) and at intervention completion (*p* = 0.001); lower levels of salty/fatty foods from baseline to intervention (*p* = 0.02); and less sugary beverages were consumed from baseline to the completion of the training period (*p* = 0.03) and at completion of the intervention (*p* = 0.04). 

Mental Health: Participants reported lower levels of anxiety and fewer microaggressions from baseline to the training period completion (*p* < 0.05), but higher levels of historical trauma (*p* = 0.02). At the completion of the intervention, participants reported lower levels of stress, anger, anxiety, PTSD, microaggression, and depression, and increased positive mental health and historical loss thoughts compared to baseline. At the six-month follow-up period, participants continued to report significantly improved positive mental health including less depression, reduced anger, and fewer reports of micro-aggressions (see Table 3).

Social/Cultural Health: During the training period participants’ Cherokee identify scores significantly increased (*p* < 0.002). At the completion of the intervention, participants reported significantly higher levels of Cherokee identity (*p* < 0.01), traditional Cherokee ways (*p* = 0.001), social support (*p* < 0.005), and Cherokee community values (*p* < 0.01). At follow-up, only Cherokee identity remained significantly increased (*p* = 0.01) from baseline (see Table 3).

## 7. Discussion

The purpose of this project was to evaluate the health effects of a popular, tribally-created leadership program designed for Indigenous youth and young adults using a prospective, pre-post-follow-up design. This was one of the first quantitative health assessments of a tribally created program in which cultural learning was the only intervention. Specifically, no Western medical health techniques, interventions, or specialists were involved in the creation or intervention of this project. The assessment was feasible with a total rate of attrition of 21% which is significantly lower than most studies, especially given multiple data collection periods over time and a small sample size. The attrition rate for this study exceeds average participant response rates (Others: 34% vs. This study: 79% at follow-up) and is comparable to other studies in which community-based participatory research principles, as well as Indigenous methods, are used [60]. Despite a small sample size, our results demonstrated statistically significantly improved physical, mental, social, and cultural health outcomes, as well as improved health behaviors during and up to six months after program completion. These results lend support to tribally-led and culturally-grounded programs to prevent and treat the health effects that Indigenous people disproportionately endure.

Physical health indicators (e.g., BMI), positive health behaviors (e.g., exercise), and self-efficacy for health behaviors approached or reached significance during or after the intervention but did not remain significant at the six-month post-test. However, diet improved during the program and including increased fruit consumption and reduced sugary beverage consumption. This is quite notable given the persistent and high rates of adolescent and adult obesity and Type II Diabetes in tribal communities [5]. While the program did not have a nutritionist, formal diet recommendations, or diet requirements, one program coordinator shared that they did ask participants to reduce their soda consumption during the program. This may speak to the importance of mentoring, positive relationships, and the positive regard that the participants had for the coordinators as discussed in a previous manuscript [34].

Many mental health indicators improved during and after this program. This is particularly impressive given the lack of any Western mental health component to this intervention. This finding is congruent with the philosophy of *culture as prevention/treatment,* which asserts that connection to or immersion in Indigenous culture has health-promoting impacts for Indigenous people and communities. In short, “culture is a social determinant of mental health/well-being for Indigenous/Native peoples” [61]. Connection and identity represent the foundation for the profound impacts of culture, which was precisely what this program worked to restore and strengthen. This is congruent with our focus group results that demonstrated increased feelings of resilience, belonging, and empowerment connected to RTR program participation [42]. These results indicate that RTR participants were significantly less lonely and reported improved social support during this program. Connections were tribally specific as Cherokee values, traditional ways, and identity increased at the completion of this program while increased Cherokee identity remained significant six months after program completion. These results align with previous works that demonstrate health and well-being improvements after cultural immersion programs in the areas of substance use [62], suicidal ideation [63], education outcomes [64], and cardiovascular risks [65,66]. For example, Suquamish and Port Gamble S’Klallam Tribes created an intervention with the goal of increasing cultural belonging to prevent substance use disorder amongst youth. That study found that a culturally grounded intervention that centers on the Coast Salish Canoe Journey resulted in increased cultural knowledge and identity, improved hope, optimism, and self-efficacy, and reduced substance use [62].

In this study, participants reported increased historical loss thoughts. However, it is not surprising that thoughts of historical loss increase when there is an increase in teaching about historically traumatic events. Researchers have argued that…

“*Implicit in the concept of historical trauma is that memories and consciousness of atrocities, such as the Trail of Tears, have multigenerational negative effects on group members, and becoming cognizant of atrocities even if this is a new awareness of the past can initiate one’s experiences of historical trauma*”.[67]

What is surprising and hopeful in this study is that historical loss was insignificant at the post-test. We hypothesize that this can be explained through the Indigenous-specific stress-disease process theory that calculates health outcomes in relation to Indigenous-specific stress and trauma, as well as protective social and cultural factors [68]. For example, although program participants are faced with increased thoughts and reminders of historical trauma through the RTR program given the brutal history of settler-colonialism in the United States, cultural enculturation appears to serve as a buffer to reduce the development of negative health outcomes consistent with previous findings [67,69,70,71].

Our previous qualitative [34] and current quantitative results indicate that participants were both angered and saddened by what they had learned surrounding Cherokee history, as well as uplifted, enlightened, and had an enhanced sense of Cherokee identity and belonging. In this study, six months after the program ended participants no longer had significantly increased rates of historical loss but had increased Cherokee identity and reduced mental health distress. Learning about social injustice informed and empowered participants to take leadership roles centering equity for Cherokees and for all people so as not to repeat racist and genocidal policies that they learned about [34]. It appears that both Indigenous and non-Indigenous students may benefit from learning about their regional Indigenous histories and current peoples to promote justice and equity for all people.

The National Indian Education Association promotes the use of Culturally and Linguistically Responsive Education as best practice for Indigenous students [72] and Indigenous researchers call on educators to implement culturally sustaining and revitalizing pedagogies to achieve educational sovereignty [73]. These interventions not only result in improved student outcomes but improved outcomes around health and well-being. For instance, in a review of Indigenous language use, results revealed that language use was associated with myriad positive health and well-being outcomes including reduced substance use, reduced suicidal ideation, reduced cigarette smoking, improved high school graduation rates, and reduced rates of diabetes [74]. Our study results are consistent with emerging research, theories, and organizations promoting positive health and well-being outcomes related to learning centered in Indigenous culture and language.

Limitations: We recognize that this study has limitations, primarily the fact that the study population is small in nature, lacks a comparison group, and is specific and generalizable to only those who participated in the Remember the Removal program. The small participant size also limited the type of analysis we were able to use which we hope to improve in the future. Modeling, for example, would give us more information about the relationship between the variables.

We employed a pre-post study design which limits our ability to draw meaningful conclusions from our study results given that participants self-selected and that we did not have a comparison group. However, this was one of the first studies to evaluate the health effects of a community-created cultural intervention in this population, therefore, it was an appropriate method to use. Future studies assessing cultural interventions should utilize culturally appropriate comparison methods to better isolate the effects of cultural intervention. For example, as described in Kaholokula et al.; “A waitlist control rather than a concurrent control group was a mandate of our community partners and stakeholders to ensure all participants were eventually offered the hula-based intervention.” [66].

Additionally, we acknowledge that the CN and EBCI Remember the Removal programs include slight differences between them, particularly participant age eligibility, as well as some departmental differences. Such differences could play a role in determining whether the results from the two programs are truly comparable or should be combined. Furthermore, the project was carried out over two years of cohorts of the RTR program, and some participants were lost to follow-up between baseline and subsequent data collection periods.

Strengths: Despite these limitations, quantitative health analysis of Indigenous community projects is novel and has demonstrated profound health effects without Western methods of health promotion. While countless Indigenous communities engage in their culture daily, this may be one of the first projects to evaluate a cultural program that was created completely within an Indigenous community absent of any academic or medical intervention [75]. Further, this study moves away from a deficit approach more commonly used in Indigenous health research and amplifies Indigenous strengths. In fact, “To this day, damage-centered narratives of Indigenous communities continue to obscure the healing, strength, and survivance that reservations can also support” [76]. The RTR program works to undo this narrative and promotes Indigenous knowledge and practices resulting in healthier, empowered, and more informed Cherokee citizens who are prepared to take leadership roles within their communities. Future projects should continue to explore unique Indigenous methods of pursuing cultural continuity and monitor health holistically and in culturally relevant ways [77].

## 8. Conclusions

This study adds to the body of research that supports culture as a critical component of positive health and well-being in Indigenous communities [78]. Therefore, it is time to elevate Indigenous knowledge and principles of health and well-being into health care delivery. In fact, Indigenous culture within medical settings is acceptable to healthcare professionals [79], is cost-effective [80], and is more effective than Western care alone for addressing health care risks amongst Indigenous populations [60,61]. Given this information coupled with the knowledge of the deep disparities that exist for Indigenous versus non-Indigenous communities on many health indicators, it is medically neglectful to not utilize Indigenous culture and community in all healthcare interventions with this population.

While cultural connectivity is a key aspect of health and well-being for Indigenous people, the suppression of Indigenous knowledge and practices remains a problem in most sectors of our society and is an international violation of sovereignty [81,82]. Until these policy-level issues are addressed, Indigenous youth and adults will remain targeted to suffer from health inequities. Future projects in this area should uplift Indigenous people and knowledge, as well as implement decolonizing and Indigenized research principles [83]. Indigenous people are citizens of sovereign nations and are experts in their lives, history, land, and health and all aspects of research must abide by these truths. Indigenous cultural knowledge and practices are indeed effective preventative and treatment measures to address chronic health diseases.

## Figures and Tables

**Figure 1 ijerph-19-08018-f001:**
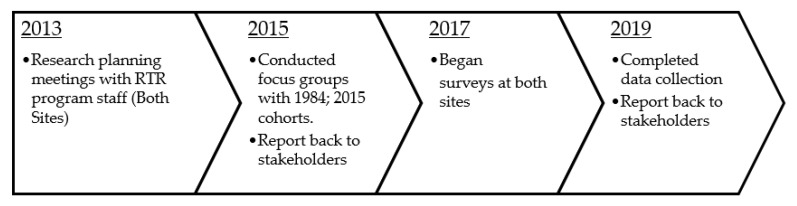
Remember the Removal Program Evaluation Project Timeline.

**Table 1 ijerph-19-08018-t001:** Participant sample size.

	Year 1	Year 2	Total
	N	N	N
Pre-training (baseline)	19	19	38
Pre-Ride	19	17	35
Post-Ride	16	15	31 *
Follow-up	16	14	30 *

Note. * One participant had missing outcome data and was excluded from the post-ride/follow-up paired *t*-test comparisons.

**Table 2 ijerph-19-08018-t002:** Sample demographics (*n* = 38).

	*n*	%
Year		
1	19	50.0
2	19	50.0
Site	14	36.8
EBCI		
Cherokee Nation	24	63.2
Age at baseline, mean (sd)	26.2	(11.5)
Gender	16	42.0
Male		
Female	22	57.9
Education	4	10.5
Less than high school		
High school or GED	10	26.3
Some college, vocational or technical training	10	26.3
College graduate	10	26.3
Advanced degree	4	10.5
Employment status	16	42.1
Full Time		
Part Time	5	13.2
Student	12	31.6
Unemployed/retired/other	5	13.2
Income	13	34.2
$19,999 or less		
$20,000 to $39,999	7	18.4
$40,000 to $69,999	9	23.7
$70,000 or more	9	23.7
Live on tribal reservation	34	89.5

**Table 3 ijerph-19-08018-t003:** Summary statistics of select outcomes.

	Base	TP1			Base	Interv			Base	Follow		
	(*n* = 35)				(*n* = 30)				(*n* = 29)			
	M	M	mDiff	*p*	M	M	mDiff	*p*	M	M	mDiff	*p*
*Physical health*												
BMI	28.18	27.83	−0.35	0.11	28.68	28.11	−0.57	0.11	28.49	28.89	+0.40	0.33
Total physical activity	2055	6893	+4838	0.001	1922	2330	+408	0.36	2194	1981	−214	0.67
Self-Efficacy	11.35	11.70	+0.35	0.09	11.26	12.00	+0.74	0.0009	11.14	11.42	+0.28	0.46
Fruit	5.74	6.66	+0.92	0.01	5.63	7.23	+1.60	0.0001	5.55	6.14	+0.59	0.12
Salty/sweet/fatty foods	6.00	5.66	−0.34	0.39	6.17	5.00	−1.17	0.02	6.17	6.31	+0.14	0.77
Sugary beverages (soda)	3.60	1.74	−1.86	0.03	4.17	2.07	−2.10	0.04	4.07	2.34	−1.73	0.06
*Mental health*												
Positive mental health	61.31	65.6	+4.29	0.07	62.5	71.17	+8.67	0.0003	61.52	66.1	+4.58	0.05
Stress	4.71	4.40	−0.31	0.20	4.73	4.03	−0.70	0.0001	4.76	4.31	−0.45	0.10
Anxiety	6.46	4.66	−1.80	0.03	6.40	3.87	−2.53	0.02	6.59	5.03	−1.56	0.14
Depression	6.69	4.89	−1.80	0.09	6.70	4.10	−2.60	0.02	6.97	3.66	−3.31	0.01
PTSD	1.20	0.94	−0.26	0.38	1.17	0.53	−0.64	0.03	1.14	0.83	−0.31	0.38
Anger	1.74	1.63	−0.11	0.10	1.76	1.60	−0.16	0.01	1.77	1.57	−0.20	0.0002
Micro-aggressions	1.07	0.93	−0.14	0.01	1.10	0.90	−0.20	0.007	1.15	0.86	−0.29	0.01
Historical trauma	3.40	3.70	+0.30	0.02	3.44	3.92	+0.48	0.0008	3.42	3.56	+0.14	0.33
*Social/cultural health*												
Social support	4.25	4.31	+0.06	0.57	4.17	4.52	+0.35	0.005	4.20	4.32	+0.12	0.39
Cherokee language	1.57	1.82	+0.25	0.09	1.66	2.00	+0.33	0.06	1.65	1.93	+0.27	0.10
Cherokee traditional ways	2.85	2.97	+0.11	0.32	2.86	3.26	+0.40	0.001	2.93	3.20	+0.27	0.07
Cherokee identity	4.32	4.52	+0.20	0.002	4.36	4.62	+0.26	0.001	4.39	4.59	+0.20	0.01
CN community values	4.42	4.52	+0.10	0.12	4.47	4.70	+0.23	0.004	4.46	4.57	+0.11	0.17

Notes. Base = Baseline; TP1 = Completion of training period; Interv = Completion of intervention (post-ride), Follow = 6 month follow-up; mDiff = mean difference (+ refers to an increase, —refers to a decrease); *p*-value based on paired t-statistic.

## Data Availability

Data is available by request.

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
