# Peer review of "The Health Effects of a Cherokee Grounded Culture and Leadership Program"

_ijerph, 2022, doi:10.3390/ijerph19138018_

Round 1
Reviewer 1 Report
The opinions discussed in this article whose title is “The health effects of a Cherokee grounded culture and leader ship program” are innovative and meaningful to practice. But there are still some issues that need to be fixed further. Details are as follows.
- Format and Written language
- The formatting of this article is not up to standard in some places, making it difficult to read and understand (e.g. some tables).
- Some of the exposition in the article is too colloquial. The authors needs to revise the presentation appropriately.
- Methods
- Please indicate the sample size in this study. Also, were there any loss of follow up by participants at various steps in the study? Has there been a change in the total number of participants?
- Please add the inclusion and exclusion criteria for participants in this study.
- As an intervention study, authors must describe in detail the process of intervention implementation, including the specific steps, items of intervention, etc.
- The authors need to present information on the scales/research instruments used in this study in more detail. Themes covered by these scales should also be explained in the background section as to why these factors were included for analysis?
- Please give details of all the statistical methods used. Also, a paired t-test was used in this study, but I am concerned that with such a small sample size, do the data meet the conditions for using a paired t-test? For example, does it satisfy the normality requirement?
- Results
- The authors need to revise the format of the tables, which are difficult to read and understand now.
- The format of the p-value in Table 2 is completely wrong.
- Some of the factors presented in the results are not described in the methods section, such as dietary status. Please explain how this data was collected.
- The results are now too weak and meaningless. To make this study more valuable, more data analysis is recommended to get richer results.
- Discussion
- The sample size of this study is too small, and the reliability of the extrapolated results is questionable.
- Reasons for loss to follow-up and the deviation in results due to it can be discussed.
- Conclusion
- The authors need to streamline the conclusion section and some of the current content should be relocated to the discussion section.
Author Response
Thank you very much for your thorough review. We believe that due to these suggestions, the manuscript is now much improved. Please see our attached tables for responses to each comment.

Reviewer 2 Report
The present study aims was to describe changes in RTR participants’ physical, emotional, social, and cultural health and well-being before, during, at completion, and six months after the completion of the program. The topic is very interesting and very relevant to improving physical and mental health conditions of the indigenous population.
The title is appropriate and indicates the main message of the paper and the abstract is well structured and clear. The objectives of the study are clearly and explicitly defined at the end of the introduction. The article provides a good and generalized background of the topic through a literature review with recent articles.
The formatting of the figures (figure 1 is too large) and tables could be improved to make them easier to read. It would be interesting to have a table with a summary of the scales used (which are quite a lot).
The work contains some self-citations that should be avoided (ex 34, 35).
The methods used are appropriate to the aims of the study and the findings of the study are properly described in the context of the published literature.
The limitations of the study are discussed and the study presents an incremental advance over previously published work and indicates directions for future studies.
Overall, the work is very well structured and the results are clearly presented.
Author Response
Thank you very much for your thorough review. We believe that due to these suggestions the manuscript is now much improved. Please see our attached tables for responses to each comment.

Round 2
Reviewer 1 Report
Thanks for inviting me again to evaluate the revised version of manuscript ijerph-1733458 entitled "The health effects of a Cherokee grounded culture and leadership program". The revised paper is well-written and is acceptable for being published.